# Optimization of adaptive filter control parameters for non-invasive fetal electrocardiogram extraction

**Radana Kahankova**©*, **Martina Mikolasova, Radek Martinek**

Department of Cybernetics and Biomedical Engineering, Faculty of Electrical Engineering and Computer Science, VSB-Technical University of Ostrava, Ostrava, Czech Republic

* radana.kahankova@vsb.cz

**Data Availability Statement:** All aECG recordings and reference data are available from the ADFECGDB database (https://physionet.org/content/adfecgdb/1.0.0/). All estimated mECG* and

## Abstract

This paper is focused on the design, implementation and verification of a novel method for the optimization of the control parameters of different hybrid systems used for non-invasive fetal electrocardiogram (fECG) extraction. The tested hybrid systems consist of two different blocks, first for maternal component estimation and second, so-called adaptive block, for maternal component suppression by means of an adaptive algorithm (AA). Herein, we tested and optimized four different AAs: Adaptive Linear Neuron (ADALINE), Standard Least Mean Squares (LMS), Sign-Error LMS, Standard Recursive Least Squares (RLS), and Fast Transversal Filter (FTF). The main criterion for optimal parameter selection was the F1 parameter. We conducted experiments using real signals from publicly available databases and those acquired by our own measurements. Our optimization method enabled us to find the corresponding optimal settings for individual adaptive block of all tested hybrid systems which improves achieved results. These improvements in turn could lead to a more accurate fetal heart rate monitoring and detection of fetal hypoxia. Consequently, our approach could offer the potential to be used in clinical practice to find optimal adaptive filter settings for extracting high quality fetal ECG signals for further processing and analysis, opening new diagnostic possibilities of non-invasive fetal electrocardiography.

## Introduction

Non-Invasive Fetal Electrocardiography (NI-fECG) is among the most promising methods for non-invasive fetal monitoring. This technique records electrical potentials from the sensors placed on maternal abdomen. These signals contain both maternal and fetal component accompanied with significant amount of noise that overlaps in time and frequency domain. In addition, the amplitude of the maternal component is usually much stronger than the fetal one [1]. This makes accurate extraction of clinically relevant features challenging. However, development of advanced signal processing methods makes NI-fECG extraction possible and thus this method could become a useful monitoring tool in the clinical practice of obstetrics and gynecology [2]. This method has the potential to emerge as an effective alternative in

aECG* recordings and other reference data are available from the ST ANNOTATIONS OF ADFECGDB DATABASE database (https://doi.org/10.21227/70cd-bw64). https://ieee-dataport.org/documents/st-annotations-adfecgdb-database.

**Funding:** R.M. received funding from European Regional Development Fund in the Research Centre of Advanced Mechatronic Systems project through the Operational Programme Research, Development and Education under Project CZ.02.1.01/0.0/0.0/16_019/0000867, R.K. and M. M. received funding from Ministry of Education of the Czech Republic under Project SP2021/32 and SP2022/34, respectively. The funders had no role in study design, data collection and analysis, decision to publish, or preparation of the manuscript.

**Competing interests:** The authors have declared that no competing interests exist.

diagnosing fetal distress to conventional method of electronic fetal monitoring, cardiotocography (CTG) [3]. The main reason is that the fECG signal carries valuable information, such as pathological states (myocardial ischemia, intrapartum hypoxia, or metabolic acidosis) manifesting as changes in the morphology of the fECG waveform, such as ST segment or QT interval, which cannot be accessed from the CTG traces because of the nature of its measurements [3].

Many different methods have been introduced for fECG signal extraction from abdominal ECG signals [2, 4, 5]. Several authors se used adaptive systems and obtained promising results. One approach is using the adaptive algorithms alone to extract fECG—thus using mECG recorded on the maternal thorax as the reference and abdominal signal as the primary input. This was applied for example by Behar et al. [6] or Martinek et al. [7]. This means using additional bioelectrodes and wires to acquire the thoracic signal that might inconvenience the patient during labor and delivery. Moreover, the quality of this signal affects the filtration results as adaptive system is generally vulnerable to the noises on system input [8, 9]. In fetal monitoring, it may be affected by the maternal motion, breathing activity or unsuitable contact of the electrode with the skin in thoracic area [2]. Due to that, it could be quite complicated to maintain the high standard of the recording in the clinical practice. Therefore, a more common approach is to combine different algorithms and creating *hybrid* extraction system. For example, Barnova et al. introduced several algorithms, including [10, 11] combining independent component analysis (ICA), empirical mode decomposition (EMD) based methods with different adaptive algorithms. Other authors, such as Jaros et al. [12, 13] or Gupta et al. [14], selected simpler approach and combined the ICA only with different adaptive algorithms.

The above mentioned works demonstrated effectiveness of the methods in fECG extraction. However, as stressed in [2], one of the important issues is system setting. In most cases, the adaptive system's setting was selected empirically. In our initial work published in [7], we paid special attention to fECG signal extraction from abdominal ECG signals using solely adaptive filtering methods. Our research demonstrated that the appropriate selection of optimal settings for adaptive systems offers the potential to significantly improve the diagnostic quality of the extracted fECG signals and consequently facilitate their clinical acceptance. The drawback of the proposed approach was the need for the *reference mECG* signal. To overcome the limitations listed above, we implemented a new approach utilizing the positives of both adaptive and non-adaptive methods, called hybrid extraction system. In our previous research, we have proposed different variants of the hybrid methods [10, 12, 15], all using only abdominal recordings and being able to estimate the maternal component using methods based on blind source separation, such as ICA or Principal Component Analysis (PCA) [16, 17]. The adaptive part of the hybrid system then estimates the fetal ECG using the outputs of the previous non-adaptive part. Similarly as in the adaptive-only approach, the need of system setting optimization emerges. Moreover, the experiments in [7] verified the hypothesis that the optimal setting of (solely) adaptive filters depends on the position of the electrodes on the mother's body. Furthermore, the results have shown that there is a *working area* (tolerance band) where AA work optimally. However, there is the problem that the recommendations introduced in [7] are based on experiments on synthetic data, so again, there is a clear need for further research utilizing real data from databases that are recognized by the scientific community.

Herein, we introduce a summary of optimization approaches for all tested hybrid methods and propose an automated system usable in clinical practice. The findings are verified on real data from open access databases and clinical practice. Since there is a lack of publicly available databases, we also verified the method using our own dataset and used the CTG trace (clinically accepted parameter reflecting current fetal health state) as a reference. This paper aims to provide recommendation for the system settings of hybrid algorithm based on the tests carried

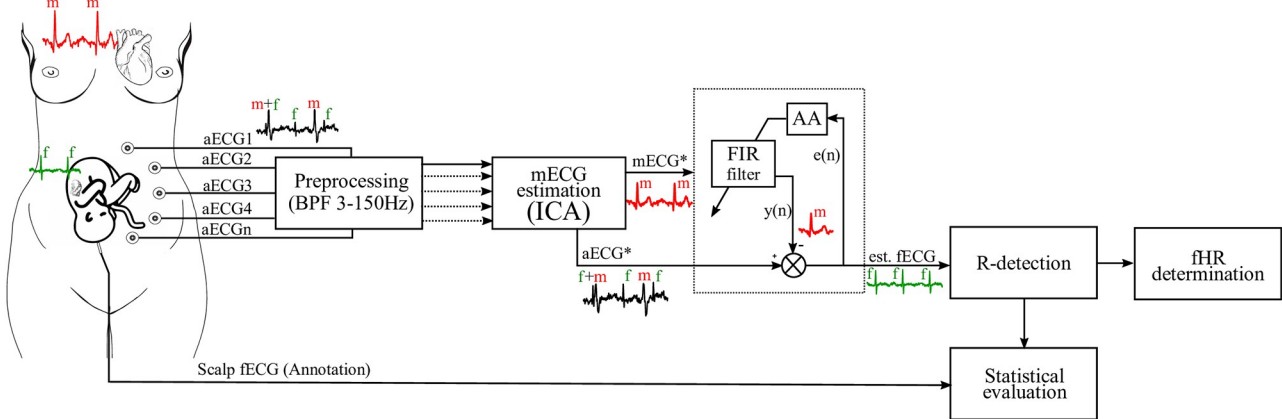

**Fig 1. A simplified block diagram of the hybrid system.** First block consists of ICA method and provides maternal reference signal ($mECG^*$) and primary input ($aECG^*$) with suppressed maternal and enhanced fetal component; second block consists of the adaptive block with adaptive algorithm (AA) which estimates the fECG signal.

on the real data from clinical practice. Using this information, an automated fECG extraction system can be developed.

## Hybrid extraction system

This subsection deals with the design and description of a hybrid system, which combines both adaptive and non-adaptive method. Fig 1 shows a block diagram of a proposed extraction system that functions as follows:

1. *Preprocessing stage*—the *aECG* signal is sensed transabdominally through electrodes labelled as $aECG_1$—$aECG_n$. The signal preprocessing phase follows, where a bandpass filter with a range of 3 –- 150 Hz is used. This filter is used to define the ECG signal band of interest and to eliminate isoline fluctuations.

2. *Non-adaptive block*—After the signal passes through the filter, the ICA method is further applied. Using this method, the input signal was divided into three components, namely the *noise*, the $mECG^*$ and $aECG^*$ components. The $aECG^*$ signal represents the fECG signal, which still contains the parent component of the signal, which, however, is partially suppressed compared to the original input signal.

3. *Adaptive block*—the reference input of this block refers to the estimated maternal component ($mECG^*$), while the primary input refers to the abdominal signal ($aECG^*$) with an enhanced fetal component and a suppressed maternal one of the same morphology as the estimated $mECG^*$ signal. The estimated fECG signal, $fECG_{est}$, is then obtained by subtracting these two components. This approach increases performance of the adaptive extraction system compared to the case where the mECG reference signal is recorded using a chest electrode.

The quality of the calculated maternal component and thus the overall extraction result of the proposed hybrid system depends on the number of aECG inputs, but also on their quality and mutual combination. The most suitable combination of electrodes entering the ICA block for the estimation of the maternal and fetal components has already been tested in [12]. After

the application of the adaptive method, the extracted $fECG_{est}$ is obtained, which ideally no longer contains the maternal component. This signal is further compared with the available reference annotations. This is followed by detection of R-peaks and determination of fHR.

The fECG extraction carried out using the adaptive block of the hybrid system is the most crucial part of the process. For the AA to function properly, its settings must be optimized. As mentioned above, AA optimization consists in selecting the most suitable setting of the filter parameters. There are several publications in which the authors deal with the optimization of an AA to improve the performance of fECG filtration (see Table 1). The optimal parameter settings can be found various methods:

- *Empirical setting*—in this case, the parameters are set based on previous experiments of the overall experience of the researcher.

- *Manual search*—this method is based on testing various settings of given method parameters and their combinations to obtain the best possible fECG extraction results. All possible combinations are can be evaluated according to various objective parameters, such as Signal-to-noise ratio (SNR), root-mean-square error (RMSE), and others [2].

- *Grid search*—in the first step of this method, a large range of values with a small step can be selected such as in [6]; in the second step, the range of parameter settings of the method is then reduced to the desired setting, when value of the selected parameter reaches its global maximum.

- *Automated search*—the method presented in [7] used 3D optimization graphs allowing automated search of optimal parameters based on the value of SNR on synthetic data.

Table 1 provides summary of publications using a fECG hybrid system with an adaptive algorithm and an optimization solution for its parameters (if mentioned). The adaptive methods used are highlighted. The most frequently used adaptive algorithms in the hybrid system include the RLS, LMS, and ANFIS methods. In general, there are not many publications on fECG signal extraction using a hybrid system using at least one adaptive algorithm. In addition, the authors do not always address the problem of adaptive optimization or choose parameters randomly. Thus, there is a potential for further research in this area.

**Table 1. Summary of state-of-the-art literature in adaptive fetal extraction systems and their optimization.**

| Author, year, source | HS for fECG extraction | Setting of AA filter parameters |
|---|---|---|
| Behar et al. (2014) [6] | LMS and RLS | Grid search (global maximum of F1) |
| Gupta et al. (2008) [14] | ICA-AFE | Not specified |
| Swarnalatha et al. (2010) [18] | WT-ANFIS | Not specified |
| | ANFIS-WT | |
| Wu et al. (2013) [19] | SWT-LMS-SSNF | Random selection of LMS filter parameters |
| Mahil et al. (2015) [20] | BC-ANN | Heuristic optimization |
| Jaros et al. (2019) [12, 13] | ICA-ANFIS-WT | Manual search (global maximum of ACC) |
| | ICA-RLS-WT | |
| Al-Sheikh et al. (2019) [21] | DWT-RI | $\lambda$ ranged from 0 to 1, $M$ set empirically. |
| | | Evaluation by Se, PPV and ACC. |
| Akhavan-Amjadi (2020) [22] | LMS-ELM | Not specified |
| Barnova et al. (2020) [10, 11] | ICA-RLS-EMD | Manual search (global maximum of ACC) |
| | ICA-RLS-EEMD | |
| Martinek et al. (2017) [7] | RLS, LMS | 3D optimization graphs (SNR, synthetic data) |

## Material and methods

Based on the latest experiments and literature review, it can be concluded that a different combination of available methods can create a hybrid system that will combine the benefits of individual methods and thus create an extraction system that will achieve far better outcomes [15]. The methods used for individual blocks of the hybrid system proposed herein are as follows:

1. *Maternal component estimation*—for mECG estimation, the ICA was used in the first block. The ICA method can highlight the fetal component in the aECG signal, but the maternal component is not sufficiently eliminated as presented in [10, 12, 16]. On the other hand, ICA is able to extract the mECG signal very accurately. The advantage of the ICA application in fECG processing is that it requires only signals from the abdominal channels and thus there is no need to record the reference mECG signal using the chest electrodes [2]. The use of solely abdominal inputs brings benefits especially for the mother since it increases her comfort and mobility. Compared to the principal component analysis method, which is also often used to estimate the maternal component, ICA achieves a more accurate estimate, and in addition, ICA produces both the *mECG** and the *aECG** signal [16].

2. *Fetal component extraction*—the AAs used for fECG signal extraction (i.e. adaptive elimination of the maternal component), were selected on the basis of literature review and study of the issue. Following algorithms were selected as most promising:

   * *Standard LMS*—this algorithm is among the most frequently used algorithms in noise cancellation systems or for system parameter identification due to its low complexity and easy implementation [23, 24]. A detailed description and implementation of the LMS algorithm can be found in [25–27].

   * *Sign-Error LMS Algorithm*—this algorithm offers faster adaptation processes and thus very fast computation which is vital in real-life applications. However, since the update mechanism is degraded by using only the sign value of the error signal compared to LMS algorithm, the steady state error increases and the convergence rate decreases [26].

   * *ADALINE*—the neural network accompanying the LMS algorithm promises higher performance. However, the computation may be slower. For more detailed information, see [27, 28].

   * *Recursive Least Squares*—the Standard RLS algorithm outperformed LMS-based algorithms in the previous test. Its advantage is its improved adjustment to the non-stationarities in the signal, however, at the cost of higher computation time. For more detailed information, see [26].

   * *Fast Transversal Filter*—this algorithm may compensate the disadvantage of the RLS algorithm since it offers comparable results at significantly lower computation time. However, this algorithm is known to suffer from numerical instability which can be compensated by optimizing its settings. For more detailed information, see [26].

### Evaluation parameters

To evaluate the quality of filtering the output signal from the proposed hybrid algorithm and compare the performance of individual methods were used in this study, objective parameters such as Accuracy (ACC), Sensitivity (Se), positive predictive value (PPV) and their harmonic

mean F1. These parameters are defined by the state of the detected fQRS complexes. Individual parameters are defined as follows:

$$Se = \frac{TP}{TP + FN},$$ (1)

$$PPV = \frac{TP}{TP + FP},$$ (2)

$$F1 = 2 \cdot \frac{PPV \cdot Se}{PPV + Se} = \frac{2 \cdot TP}{2 \cdot TP + FP + FN},$$ (3)

where TP—True Positive indicates the correct detection of QRS complexes of the fetus, i.e. that the method detects the fQRS complex that actually occurs in the signal. FP—False Positive indicates incorrect detection, the method detects an fQRS complex that does not occur in the signal. FN—False Negative indicates missed fQRS, the method does not detect the fQRS complex that is in the signal.

## Dataset

The data used in the optimization process were collected in clinical conditions as part of research projects at the Department of Obstetrics and Gynecology of the Medical University of Silesia in Katowice, Poland. The research was approved by the competent University Bioethics Committee (Commission approval number NN-013-345/02), and by each of the hospitalized patients. Subjects read the approved consent form and gave written informed consent to participate in the study. The dataset used can be divided into three groups (Dataset A, B, and C):

- *Dataset A*—contains ten real records of pregnant women with the designation r01-r10. These are recordings taken during childbirth using four abdominal electrodes placed around the navel, supplemented by a reference signal from the scalp electrode [29, 30]. Gestational age of the fetus is between 38 and 41 weeks. For recordings r01, r04, r07, r08, and r10, the sampling frequency of 1 kHz was used, for the other five recordings the sampling frequency was 500 Hz. An example of the aECG signals for recording r01, including the electrode placement display, is shown in Fig 2. Abdominal ECG signals were sensed using four electrodes labelled V1, V2, V3, and V4. The scalp electrode is denoted as V0 and the electrode N represents the active ground.

- *Dataset B* contains data from the Computing Cardiology Challenge 2013 database available on the PhysioNet website. All records contain 4 abdominal signals and one signal containing the positions of fetal QRS complexes. Since the Challenge dataset contains both real and synthetic signals from different databases, recorded/generated using different instrumentations and methods, results have an inhomogeneous distribution. Moreover, as mentioned in [31, 32], for some signals, such as a02, a09, a18, or a29, due to the low signal quality, it is not possible to correctly identify the fetal beats obtaining unreliable results. Such a database is therefore not a real representation of one's algorithm performance. However, since many authors tested their systems on this database, it is needed for an objective comparison of the results.

- *Dataset C* contains real aECG signals obtained from a pregnant volunteer in the 34th week of pregnancy. The dataset included 6 simultaneously measured aECG signals and a reference CTG signal. The total recording time is 60 minutes and the sampling frequency is 600 Hz. This recording consists of 3 parts acquired during the same day, however, each 20 minutes differ in the activity of the fetus. In the first stage, the fetus was asleep and its heart rate was

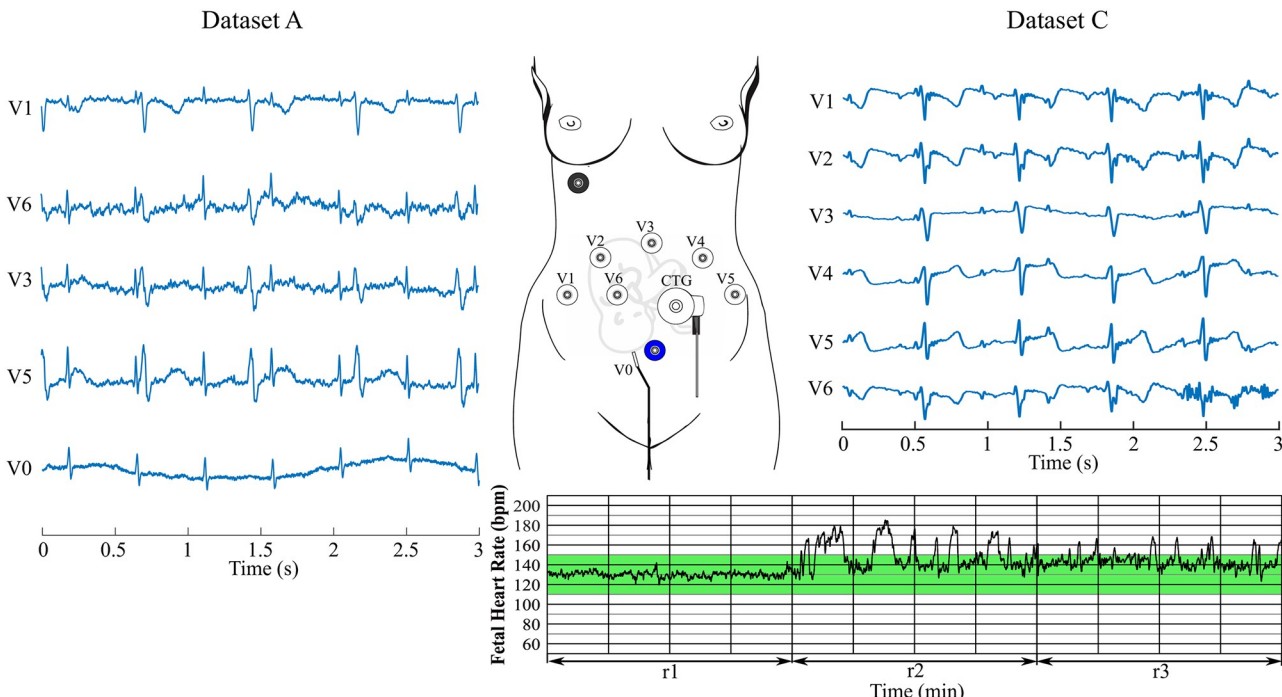

**Fig 2. Electrode placement and examples of real signals from the datasets.** A (ADFECGDB, abdominal ECGs and direct fECG from FSE) and C (Own data—abdominal ECGs and CTG trace).

stable and quite low (±130 bpm). The following stage consists of data when the fetus woke up and started moving which resulted in significant rise in fHR (up to 180 bpm) and caused motion artifacts in the signal. Finally, in the last 20 minutes, the fetus was awake but was not moving, thus both the signal and fHR were more stable (140–160 bpm).

## Results

In this section, we present experimental results obtained by individual hybrid methods on real data. As mentioned above, a total of ten data sets labeled r01–r10 and five different combinations of adaptive and non-adaptive methods (ICA-LMS, ICA-SeLMS, ICA-ADALINE, ICA-RLS and ICA-FTF) were tested. The experimental procedure was identical for each dataset and for each hybrid method.

In the first iteration, it was necessary to find the optimal filter settings for each of the tested methods and each record. Optimization 3D graphs were used for this purpose. After finding a suitable setting, a statistical evaluation of the achieved results was performed. In the following subchapters, we describe optimization graphs and optimization results, summarize all the resulting values of statistical parameters for all records and each of the tested hybrid methods, and finally illustrate several examples of comparison of input and output signals, including evaluation of fetal heart rate variability.

### Optimization

Before the actual extraction of the fECG signal, it further necessary to find the optimal setting of the adaptive filter for each of the tested methods. The parameters of individual algorithms differ, and their settings and values are different depending on the type of the signal or use.

The effect of the filter settings was tested for the selected adaptive filtering. Most of the tested algorithms are part of the Matlab Digital Signal Processing Systems Toolbox, which ensures the reproducibility of the achieved results. Most of the selected adaptive algorithms have two crucial parameters to be selected and tuned, but there are cases, where there are 3 and more important parameters to be optimized. For more details, see [2]. The parameters of the algorithms tested herein are as follows:

1. *Least Mean Squares algorithms*

    a.  Standard LMS

        - filter length $M$ (or filter order $N$),

        - step size $\mu$,

    b.  Sign-Error LMS

        - filter length $M$ (or filter order $N$),

        - step size $\mu$,

2. *Recursive Least Squares (RLS) algorithm*

    - filter length $M$ (or filter order $N$),

    - forgetting factor $\lambda$,

3. *Fast Transversal filter*

    - filter length $M$ (or filter order $N$),

    - forgetting factor $\lambda$,

4. *Adaptive Linear Neuron*

    - learning rate $\eta$,

    - input space $p$.

The importance of the optimization process in finding the suitable system setting is shown in Fig 3. The figure shows that the performance of the same system varies based on the filter settings and may result in increase of false positive values (marked as red circles) of detected R waves leading to inaccurate diagnosis of fetal health state.

The evaluation of the correct setting of parameters influencing the quality of filtration was assessed on the basis of optimization graphs showing the dependence of the value of tested parameters of the given adaptive method and the size of the statistical parameter F1. The optimization process of the ADALINE algorithm will be presented in detail as an example. As mentioned above, the extraction system using the ADALINE has two main parameters that must be optimized: learning rate $\eta$ and input space $p$.

In the first iteration, the authors tested a wide range of these parameters, which made it possible to find the working area and extremes of the algorithm. These experiments showed that for further tests the setting of parameters in the range $\eta \leq 0.1$ and $p \leq 100$ is sufficient. The experiments were performed as follows: for each record, the value of the F1 parameter was calculated for the given range and window length of 1 minute; as the optimal solution, the value determined as the global maximum of the searched area was chosen; optimizing 3D graphs are available for each record.

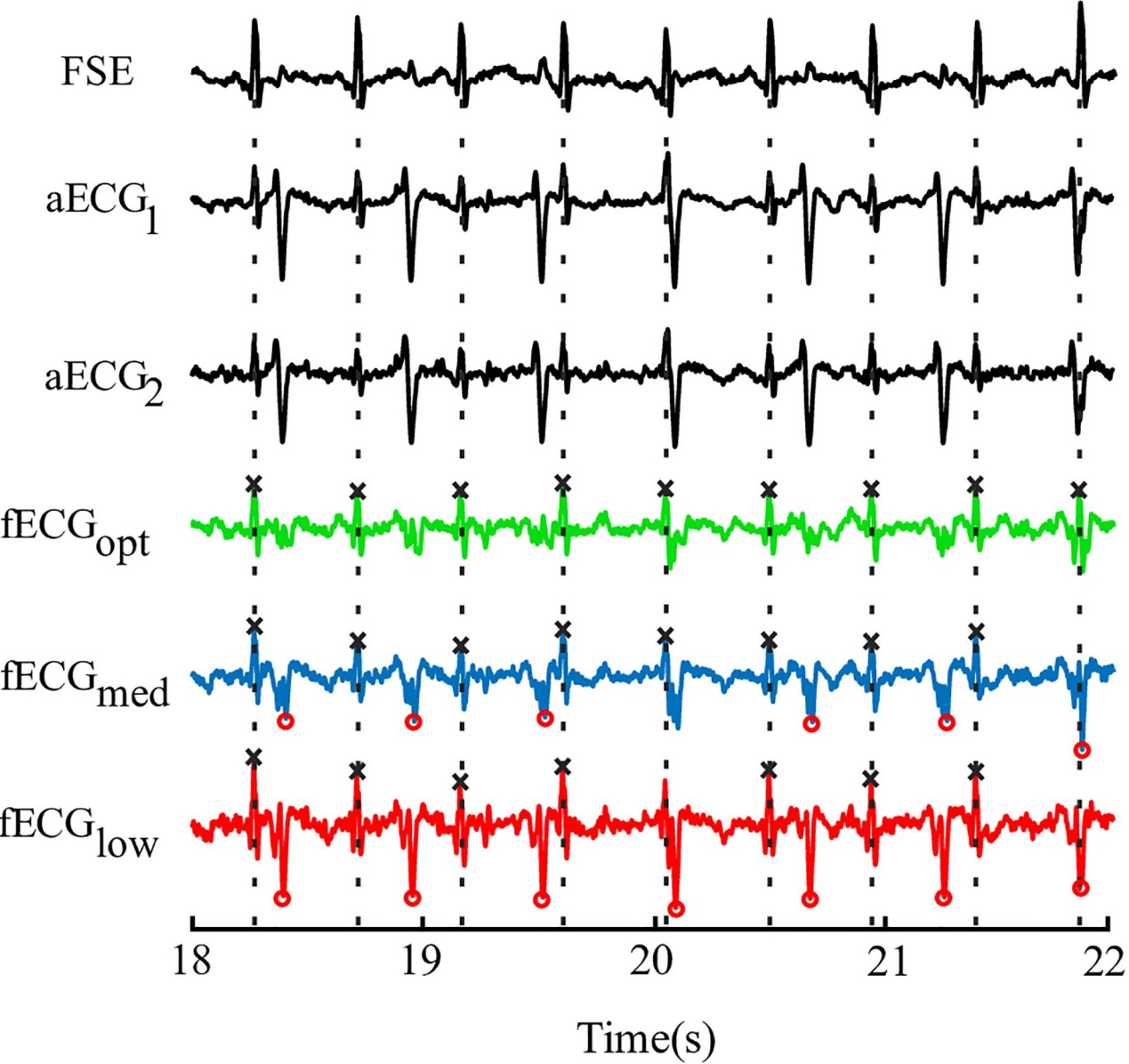

**Fig 3. Illustration of the effect optimization of adaptive system control parameters.** Examples of the input signals, reference FSE signal, and three outputs of the extraction system: optimal (green), medium (blue), and low (red) quality of the filter setting).

Table 2 summarizes the optimal parameter settings for each recording and each tested method. Figs 4–6 provide examples of the 3D optimization graphs of all tested methods and the same recordings (r01) showing the dependence of the size of the parameters on the resulting value of the parameter F1, which is encoded in color in the graphs. The color of the surface varies according to the F1 value. The color bar shows how the data values correspond to the colors in the colormap. The redder the graph area, the better the setting of all selected parameters.

- *ADALINE*—in Fig 4, it can be noticed that the filter is stable and efficient in most of the operation area, the highest performance was reached for $p < 40$. The efficacy of the filter

**Table 2. Optimal extraction system settings for each recording and each tested method.**

| | Param. | r01 | r02 | r03 | r04 | r05 | r06 | r07 | r08 | r09 | r10 |
|---|---|---|---|---|---|---|---|---|---|---|---|
| ADALINE | $\eta$ | 0.001 | 0.005 | 0.007 | 0.001 | 0.001 | 0.052 | 0.001 | 0.014-0.041 | 0.24-0.076 | 0.003 |
| | $p$ | 29-34 | 59 | 31 | 48 | 82 | 8 | 40 | 5 | 17 | 46 |
| | F1 (%) | 99.77 | 98.43 | 98.12 | 79.28 | 99.61 | 93.15 | 79.38 | 99.85 | 99.08 | 93.95 |
| LMS | $\mu$ | 0.003-0.025 | 0.097-0.1 | 0.1 | 0.004 | 0.01-0.03 | 0.1 | 0.003 | 0.007-0.009 | 0.049-0.083 | 0.012-0.030 |
| | $M$ | 30 | 19-23 | 22 | 46 | 18-41 | 1 | 46 | 46-100 | 63-67 | 54-56 |
| | F1 (%) | 99.77 | 98.35 | 91.68 | 75.52 | 99.61 | 93.27 | 77.61 | 99.92 | 98.70 | 94.32 |
| SE-LMS | $\mu$ | 0.001-0.007 | 0.035-0.054 | 0.033 | 0.07 | 0.003 | 0.038 | 0.001 | 0.009-0.1 | 0.05. 0.051 | 0.007 |
| | $M$ | 26-37 | 36-38 | 23 | 19 | 18 | 16 | 45 | 1-5 | 19. 22 | 55 |
| | F1 (%) | 99.85 | 98.58 | 86.71 | 76.53 | 99.61 | 92.28 | 74.139 | 99.85 | 98.85 | 91.63 |
| RLS | $\lambda$ | 0.997-0.999 | 0.998 | 1 | 1 | 1 | 0.997-0.998 | 1 | 0.998-0.999 | 0.999 | 1 |
| | $M$ | 2-18 | 13 | 44 | 34 | 33-97 | 20. 21 | 44 | 3. 5. 6 | 16 | 93 |
| | F1 (%) | 99.69 | 98.87 | 98.18 | 76.23 | 99.46 | 92.89 | 80.16 | 99.85 | 99.08 | 93.45 |
| FTF | $\lambda$ | 0.99-1 | 0.999 | 0.980 | 1 | 0.997 | 0.997 | 1 | 0.998 | 0.997 | 0.999 |
| | $M$ | 24-100 | 12 | 10 | 48 | 17-18 | 1 | 47 | 30. 31 | 14 | 45 |
| | F1 (%) | 99.77 | 98.80 | 86.90 | 73.13 | 99.61 | 50.51 | 72.02 | 99.92 | 98.85 | 93.20 |

decreases steeply for area defined as $\eta \geq 0.07$ and $p \geq 60$. This figure also shows the examples of the estimated fECG signals with selected filter settings to demonstrate the importance of the proper filter settings. If the filter is not optimized, it can lead to unwanted suppression of the fetal component (especially fetal R waves) in the estimated signal, which are then not detected leading to increase of the peaks classified as and false negative (see Fig 4 examples b and c).

- *RLS-based systems*—one can notice that the 3D optimization graphs in Fig 6 follow the same trends. In these cases, the algorithms share the same operating area ($M \in (14, 40)$ and $\lambda > 0.99$), and also achieve nearly the same results (local maxima $F1_{RLS} = 99.69\%$, $F1_{FTF} = 99.77\%$). The advantage of using FTF over the standard RLS algorithm is its computational speed, which is an important factor for the implementation of real-life applications. Our experiments showed that the computational time increases with the higher values of the filter length. However, for the standard RLS algorithm the increase is steep for $M > 80$ whereas

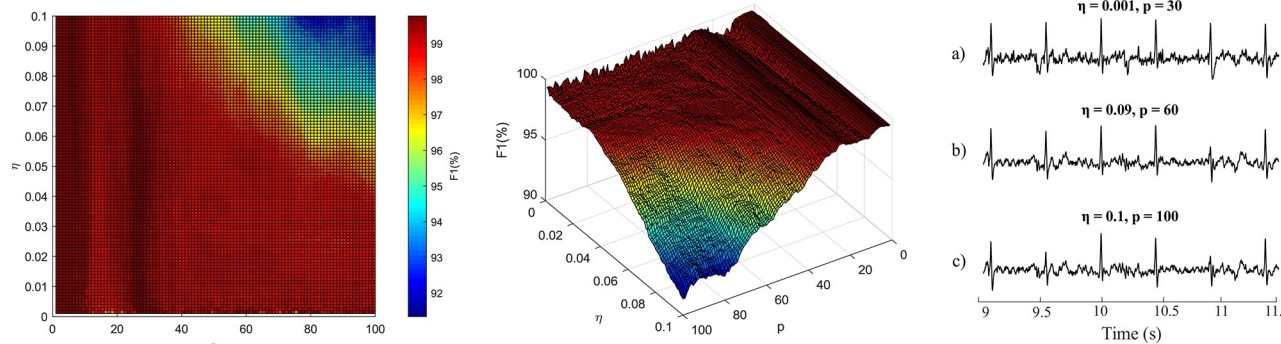

**Fig 4. Optimization of adaptive system control parameters for r01.** 3D graph showing the influence of parameter $p$ and $\eta$ on the quality of the filtration; the top-down view of the 3D graph; down right: examples of the estimated fECG signals obtained with selected filter settings: a) $\eta = 0.001$, $p = 30$, $F1 = 99.77\%$ (643 TP, 2 FP, 1 FN), b) $\eta = 0.09$, $p = 60$, $F1 = 95.8\%$ (593 TP, 1 FP, 51 FN), c) $\eta = 0.1$, $p = 100$, $F1 = 43.43\%$ (543 TP, 2 FP, 101 FN).

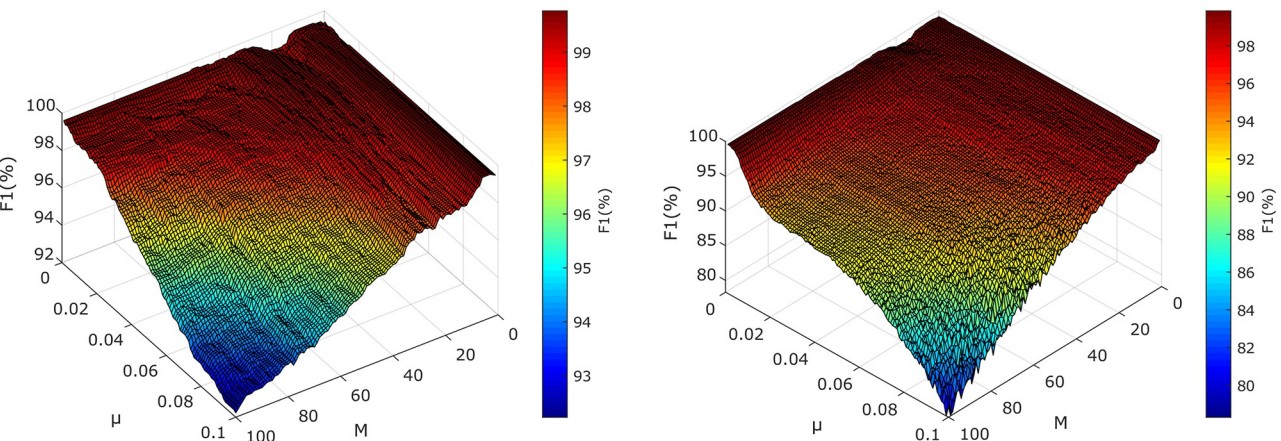

**Fig 5. Comparison of the results of LMS-based systems for recording r01, $\mu \in (0, 0.1)$, $M \in (0, 100)$.** From left to right: a) System using Standard LMS algorithm; b) System using Sign Error LMS algorithm.

for FTF-based system, the increase is gradual. At the same time, it is important to realize that the operating area for majority of the tested algorithms contains lower values of the filter length. Additionally, since the update mechanism of the FTF algorithm suffers from numerical instability, which makes it less reliable than the standard RLS algorithm.

- *LMS-based systems*—Fig 5 shows the influence of filter settings on the filtration quality for the recording r01. Both 3D optimization graphs follow the same trend; both systems are stable and effective within most of the tested range, while the global maxima ($F1_{LMS}$ = 99.77%, $F1_{SE-LMS}$ = 99.84% were found in the range $\mu \in (0.003, 0.025)$ and $M = 30$ for LMS-based system and $\mu \in (0.001, 0.007)$, $M \in (26, 37)$ for the SE-LMS based system. The advantage of the Sign-Error LMS algorithm is the speed of adaptation processes allowing very fast computation which is vital in real-life applications. However, since the update mechanism is degraded by using only the sign value of the error signal, the steady state error may increase, while the convergence rate decreases. Nevertheless, the results imply that for the fECG

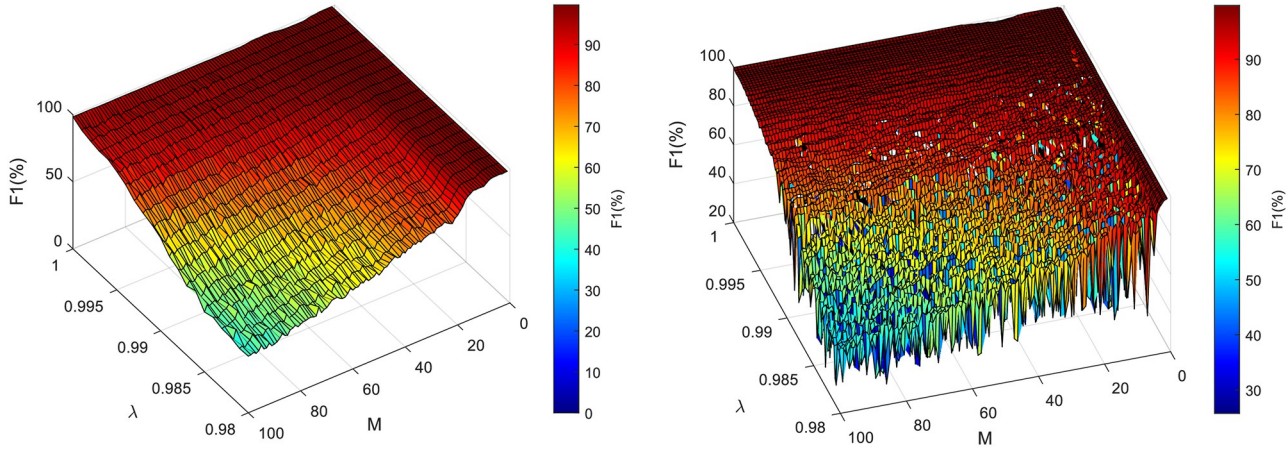

**Fig 6. Comparison of the results of FTF-based systems for recording r01, $\lambda \in (0.9, 1)$, $M \in (0, 100)$.** From left to right: a) System using Standard RLS algorithm; b) System using FTF algorithm.

**Table 3. Results of the experiments on dataset A.**

|  | ADALINE | | | LMS | | | SE-LMS | | | RLS | | | FTF | | |
|---|---|---|---|---|---|---|---|---|---|---|---|---|---|---|---|
|  | F1(%) | Se(%) | PPV(%) | F1(%) | Se(%) | PPV(%) | F1(%) | Se(%) | PPV(%) | F1(%) | Se(%) | PPV(%) | F1(%) | Se(%) | PPV(%) |
| r01 | 99.77 | 99.69 | 99.84 | 99.77 | 99.84 | 99.69 | 99.84 | 99.84 | 99.84 | 99.69 | 99.84 | 99.54 | 99.77 | 99.84 | 99.69 |
| r02 | 98.43 | 99.70 | 97.19 | 98.35 | 99.55 | 97.19 | 98.58 | 99.70 | 97.48 | 98.87 | 99.70 | 98.06 | 98.80 | 99.55 | 98.06 |
| r03 | 98.10 | 98.39 | 97.82 | 91.68 | 86.99 | 96.91 | 86.71 | 85.38 | 88.08 | 98.18 | 98.39 | 97.96 | 86.90 | 82.89 | 91.30 |
| r04 | 79.28 | 79.59 | 78.96 | 75.52 | 74.68 | 76.38 | 76.53 | 77.37 | 75.70 | 76.23 | 82.44 | 70.88 | 73.13 | 67.41 | 79.92 |
| r05 | 99.61 | 99.84 | 99.38 | 99.61 | 99.84 | 99.38 | 99.61 | 99.84 | 99.38 | 99.46 | 99.84 | 99.08 | 99.61 | 99.85 | 99.38 |
| r06 | 93.14 | 94.81 | 91.55 | 93.27 | 92.58 | 93.98 | 92.28 | 89.61 | 95.12 | 92.89 | 93.18 | 92.63 | 50.51 | 40.06 | 68.35 |
| r07 | 79.37 | 80.38 | 78.38 | 77.61 | 72.41 | 83.61 | 74.14 | 66.99 | 83.00 | 80.16 | 77.67 | 82.82 | 72.02 | 62.20 | 85.53 |
| r08 | 99.85 | 100 | 99.69 | 99.92 | 99.85 | 100 | 99.85 | 100 | 99.69 | 99.85 | 100 | 99.69 | 99.92 | 99.85 | 100 |
| r09 | 99.08 | 98.63 | 99.54 | 98.70 | 98.02 | 99.38 | 98.85 | 98.48 | 99.23 | 99.08 | 98.63 | 99.53 | 98.85 | 98.17 | 99.54 |
| r10 | 93.95 | 97.49 | 90.66 | 94.32 | 96.39 | 92.33 | 91.63 | 91.99 | 91.28 | 93.45 | 95.13 | 91.82 | 93.20 | 93.56 | 92.83 |

extraction, the SE-LMS algorithm does not perform significantly faster than the standard LMS algorithm.

## Statistical evaluation

After obtaining the optimal setting for each record and each method, a further statistical evaluation of the achieved results could can be carried out. When evaluating the quality of filtration, the greatest emphasis was placed on the value of parameter F1, expressing the total accuracy of the proposed system. The resulting value of the F1 parameter should be as high as possible. If the value of $F1 > 90\%$, the proposed system can be considered effective. If the size of the F1 parameter reaches 80%, then the results are considered satisfactory. Table 3 shows the summary of results for all of the tested algorithms and all recordings.

According to Table 3, we can state that the application of the LMS method shows excellent performance for most of the tested datasets. According to parameter F1, the accuracy of more than 99% was achieved for records r01, r05, and r08.

Outstanding results were also obtained for records r02 and r09, where the value of F1 is over 98%. According to the PPV parameter, an accuracy of over 95% was achieved for records other than r01–r03, as well as for records r05, r08 and r09. Accuracy over 80% was achieved with the PPV parameter achieved for records r06 and r10. According to the Se parameter, an accuracy of more than 80% was achieved for all but the records r04 and r07. The worst results were achieved for records r04 and r07, where the value of the parameter F1 did not exceed the limit of 80%.

Based on the results contained in Table 3, it can be stated that the application of the ADALINE method shows excellent performance for most of the tested datasets. The best results were obtained with records r01, r05, r08, and r09, where the value of the parameter F1 exceeds 99%. These records also have excellent results for the PPV and Se parameters, where in both cases the accuracy of 95% is exceeded. The poor results when testing the ADALINE method were achieved with records r04 and r07, where the value of the F1 parameter is very close to 80%.

As for the results of Sign-Error LMS method, the best results summarized in Table 3 were achieved for records r01, r02, r05, r08 and r09. Furthermore, satisfactory results ($F1 > 80\%$) were obtained for records r03, r06 and r10. As in the case of ADALINE-based hybrid system, the worst results were obtained for records r04 and r07, where the value of F1 is around 75%.

The extraction quality was evaluated using a combination of ICA-RLS and ICA-FTF methods. In the case of the RLS method, excellent results ($F1 > 95\%$) were obtained for records r01-r03, r05, r08, and r09. Furthermore, good results were obtained with an accuracy of over 80% of the F1 parameter for records r06, r07 and r10. The worst result was achieved in r04. For the FTF method, excellent results were also obtained for records r01, r05 and r08 ($F1 > 99\%$). On the contrary, the worst results, where the value of the parameter F1 did not exceed 75%, were achieved for records r04, r06 and r07. For the r06 record, the accuracy of the F1 parameter was even only 51%.

Finally, to be able to compare the results with other authors, the extraction of fECG was also performed on Dataset B. Again, the most suitable combination of electrodes entering the ICA block was selected, then the filtration parameters were optimized for each of the tested adaptive algorithms and finally the detection of R-peaks extracted fECG signal using a CWT-based detector and selected statistical parameters were calculated. Table 4 summarizes the results of statistical parameters for all tested algorithms.

## Verification of own data

The aECG signals were sensed using six electrodes (aECG1-aECG6) placed on the mother's abdomen. From these six measured aECG signals, 57 possible combinations were subsequently created to enter the ICA algorithm, from which the one that achieved the best results was subsequently selected. After the optimal combination of electrodes was found, the proposed hybrid algorithm with different blocks of the adaptive system was tested.

The results show that the proposed hybrid system is able to extract the fECG signal accurately enough to determine the fHR trace of the same quality as the CTG method. Fig 7 shows the fHR traces determined using the NI-fECG signal extracted by the hybrid system for all tested adaptive extraction blocks. It can be noticed that the fHR traces follow the same trend as the CTG trace. However, there are some deviations from the CTG reference (for example in case of LMS, SE-LSM and RLS algorithms) caused by the inaccurate extraction. On the other hand, the ADALINE-based extraction system can be considered as the most accurate one regarding the CTG reference, the fHR trace deviate minimally (±10 BPM). Nevertheless, one should keep in mind that CTG signal undergoes several processing phases, including autocorrelation and averaging, therefore the resulting fHR traces will hardly ever have the exact morphology.

## Discussion

This part is devoted to summarizing and discussing the achieved results, and comparing them with those achieved by other authors. Furthermore, we present remaining challenges and future directions to increase the feasibility of NI-fECG monitoring into clinical practice.

The performance of the hybrid system varied depending on the adaptive algorithm used in the fECG extraction block. The LMS algorithm was used in three different versions of extraction systems—the standard LMS; signal error-based system SE-LMS, and ADALINE. Of these three types of extraction systems, the ADALINE-based method shows the best performance for all three datasets.

The results of experiments performed on data from the ADFECGDB database show that the proposed hybrid system using the ADALINE, LMS, and RLS methods in combination with the ICA method shows excellent performance in most of the tested data sets and is able to determine fHR with relatively high accuracy. The main prerequisite for achieving quality results is the correct estimation of mEKG using the ICA method. The estimation of individual components is based on statistical methods that lead to unpredictable results. In Fig 8,

**Table 4. Results of the experiments on dataset B from the challenge 2013 database.**

| | LMS | | | SE LMS | | | ADALINE | | | RLS | | | FTF | | |
|---|---|---|---|---|---|---|---|---|---|---|---|---|---|---|---|
| | Se(%) | PPV(%) | F1(%) | Se(%) | PPV(%) | F1(%) | Se(%) | PPV(%) | F1(%) | Se(%) | PPV(%) | F1(%) | Se(%) | PPV(%) | F1(%) |
| a01 | 94.48 | 94.48 | 94.48 | 93.13 | 95.75 | 94.41 | 95.17 | 93.24 | 94.19 | 89.66 | 90.28 | 89.97 | 81.21 | 92.41 | 89.63 |
| a02 | 18.75 | 24.45 | 21.05 | 42.56 | 53.97 | 47.55 | 27.53 | 51.16 | 35.39 | 40.02 | 48.86 | 43.99 | 40.78 | 29.91 | 34.23 |
| a03 | 92.19 | 91.47 | 91.83 | 96.44 | 93.89 | 94.98 | 95.31 | 88.41 | 91.73 | 94.53 | 92.37 | 93.44 | 63.64 | 58.82 | 58.82 |
| a04 | 100 | 100 | 100 | 100 | 100 | 100 | 100 | 100 | 100 | 100 | 100 | 100 | 100 | 98.47 | 99.23 |
| a05 | 100 | 100 | 100 | 100 | 100 | 100 | 100 | 100 | 100 | 100 | 100 | 100 | 100 | 99.23 | 99.61 |
| a06 | 56.25 | 62.94 | 59.41 | 56.88 | 63.19 | 59.87 | 56.88 | 64.09 | 61.13 | 57.53 | 62.59 | 59.94 | 55.63 | 62.24 | 58.75 |
| a07 | 56.92 | 44.31 | 49.83 | 60.76 | 46.25 | 52.49 | 57.69 | 42.86 | 49.18 | 58.46 | 44.44 | 50.56 | 60.34 | 42.62 | 49.84 |
| a08 | 100 | 100 | 100 | 100 | 100 | 100 | 100 | 100 | 100 | 100 | 100 | 100 | 99.22 | 100 | 99.60 |
| a09 | 41.54 | 39.13 | 40.29 | 46.15 | 43.83 | 44.94 | 36.92 | 36.36 | 36.64 | 51.54 | 40.85 | 45.58 | 43.85 | 28.22 | 34.34 |
| a10 | 88.67 | 92.22 | 90.06 | 88.34 | 89.54 | 88.76 | 87.43 | 91.07 | 89.21 | 82.29 | 84.71 | 83.48 | 75.43 | 80.98 | 78.11 |
| a11 | 16.43 | 52.27 | 25 | 52.86 | 57.36 | 55.02 | 54.29 | 56.31 | 56.73 | 56.43 | 64.23 | 60.08 | 40.71 | 30.32 | 34.76 |
| a12 | 85.51 | 80.27 | 82.81 | 70.32 | 75.19 | 72.66 | 88.41 | 88.41 | 90.93 | 93.48 | 93.48 | 93.48 | 79.71 | 83.33 | 81.48 |
| a13 | 70.64 | 89.03 | 78.76 | 56.35 | 70.38 | 62.56 | 88.09 | 82.84 | 91.02 | 88.19 | 86.05 | 87.06 | 62.77 | 50.22 | 55.63 |
| a14 | 84.55 | 77.04 | 80.62 | 76.42 | 72.87 | 74.66 | 90.24 | 79.86 | 84.73 | 91.87 | 80.14 | 85.16 | 76.42 | 75.27 | 75.81 |
| a15 | 100 | 100 | 100 | 100 | 100 | 100 | 99.25 | 99.25 | 100 | 100 | 100 | 100 | 99.25 | 100 | 99.63 |
| a16 | 14.62 | 21.11 | 17.27 | 53.85 | 40.94 | 46.51 | 43.85 | 36.08 | 40.97 | 50.34 | 39.63 | 44.22 | 59.23 | 36.84 | 45.43 |
| a17 | 100 | 100 | 100 | 100 | 99.25 | 99.62 | 100 | 98.51 | 99.62 | 98.49 | 97.74 | 98.11 | 96.97 | 92.75 | 94.82 |
| a18 | 18.67 | 22.76 | 20.51 | 18.67 | 23.53 | 20.82 | 18.45 | 23.08 | 21.05 | 31.33 | 29.01 | 30.13 | 34.96 | 29.31 | 31.48 |
| a19 | 92.91 | 92.91 | 92.91 | 91.14 | 80.47 | 70.78 | 85.04 | 78.83 | 92.97 | 96.06 | 91.73 | 93.85 | 57.48 | 57.94 | 57.71 |
| a20 | 84.73 | 86.72 | 85.71 | 69.47 | 67.91 | 68.67 | 89.31 | 86.68 | 87.97 | 79.39 | 81.89 | 80.62 | 49.62 | 40.88 | 44.83 |
| a21 | 77.93 | 76.35 | 77.13 | 79.31 | 82.73 | 80.99 | 76.55 | 76.03 | 76.23 | 75.86 | 72.85 | 74.32 | 74.48 | 73.47 | 73.97 |
| a22 | 100 | 100 | 100 | 100 | 100 | 100 | 100 | 100 | 100 | 100 | 99.21 | 99.61 | 100 | 98.44 | 99.21 |
| a23 | 56.35 | 65.74 | 60.68 | 62.72 | 66.39 | 64.49 | 91.27 | 83.33 | 91.63 | 77.78 | 70.58 | 73.96 | 58.73 | 45.68 | 51.39 |
| a24 | 91.06 | 83.58 | 87.16 | 94.31 | 87.88 | 90.98 | 91.87 | 75.84 | 86.25 | 91.06 | 84.21 | 87.56 | 63.42 | 54.93 | 58.87 |
| a25 | 87.25 | 90.08 | 88.62 | 71.21 | 76.72 | 73.86 | 96.88 | 89.55 | 92.66 | 85.64 | 86.99 | 86.29 | 45.61 | 51.35 | 48.31 |
| a28 | 86.23 | 97.96 | 91.72 | 86.23 | 96.64 | 91.14 | 87.43 | 97.98 | 92.41 | 92.81 | 97.48 | 95.09 | 79.64 | 93.01 | 85.81 |
| a35 | 91.41 | 92.55 | 91.98 | 95.09 | 94.51 | 94.8 | 92.64 | 92.64 | 92.64 | 76.07 | 85.52 | 80.52 | 80.37 | 86.76 | 83.44 |
| a36 | 93.49 | 96.34 | 94.90 | 79.88 | 85.44 | 82.57 | 91.72 | 95.68 | 93.66 | 89.35 | 92.64 | 90.96 | 76.33 | 82.17 | 79,14 |
| a44 | 98.16 | 100 | 99.07 | 98.16 | 100 | 99.07 | 96.32 | 99.38 | 97.82 | 98.16 | 100 | 99.07 | 93.87 | 98.71 | 96.23 |
| a49 | 100 | 95.48 | 97.69 | 95.95 | 98.61 | 97.26 | 100 | 100 | 100 | 98.65 | 98.65 | 98.65 | 94.59 | 93.96 | 94.28 |
| a55 | 65.04 | 84.55 | 73.52 | 63.64 | 66.91 | 65.23 | 81.82 | 81.82 | 81.82 | 66.02 | 83.52 | 73.48 | 65.04 | 49.21 | 56.02 |
| a61 | 97.86 | 100 | 98.92 | 99.29 | 99.29 | 99.29 | 97.86 | 100 | 98.92 | 100 | 97.99 | 98.94 | 97.86 | 97.16 | 97.51 |
| a62 | 70.83 | 76.12 | 73.38 | 65.28 | 68.83 | 65.05 | 92.36 | 89.87 | 91.10 | 65.28 | 73.44 | 69.12 | 55.56 | 74.07 | 63.49 |
| a65 | 76.39 | 97.35 | 85.61 | 70.83 | 91.07 | 79.69 | 87.51 | 99.21 | 92.99 | 90.28 | 97.75 | 93.86 | 76.39 | 56.12 | 64.71 |
| a66 | 100 | 50.43 | 66.67 | 100 | 51.78 | 66.67 | 100 | 50.53 | 66.67 | 100 | 50.89 | 66.67 | 100 | 55.98 | 66.67 |
| a67 | 91.56 | 93.37 | 92.46 | 88.96 | 93.22 | 91.03 | 92.86 | 95.33 | 94.08 | 86.36 | 93.66 | 89.87 | 64.29 | 77.34 | 70.21 |
| a69 | 87.25 | 87.84 | 87.54 | 81.21 | 93.08 | 86.74 | 89.26 | 89.26 | 89.26 | 85.24 | 96.95 | 90.71 | 72.48 | 77.72 | 75.64 |
| a70 | 92.91 | 94.25 | 93.57 | 95.75 | 93.17 | 94.41 | 92.27 | 91.55 | 91.87 | 82.98 | 90.67 | 86.35 | 63.83 | 60.01 | 61.86 |
| a72 | 91.62 | 95.63 | 93.59 | 93.41 | 97.53 | 95.41 | 95.21 | 98.76 | 96.95 | 95.81 | 98.16 | 96.97 | 83.23 | 78.09 | 80.58 |

examples of aECG signals from the ADFECGDB database are shown along with signals estimated using ICA. These signals were further used as primary ($aECG^*$) and reference ($mECG^*$) inputs of the adaptive algorithm. The performance of the adaptive algorithm strongly correlates with the quality of its inputs, especially with the reference signal $mECG^*$.

A total of 6 tested recordings were selected for illustration. Recordings r01, r02 and r08 that showed the best results ($F1 > 95\%$) in fECG extraction; recordings r04, r06, and r07, which

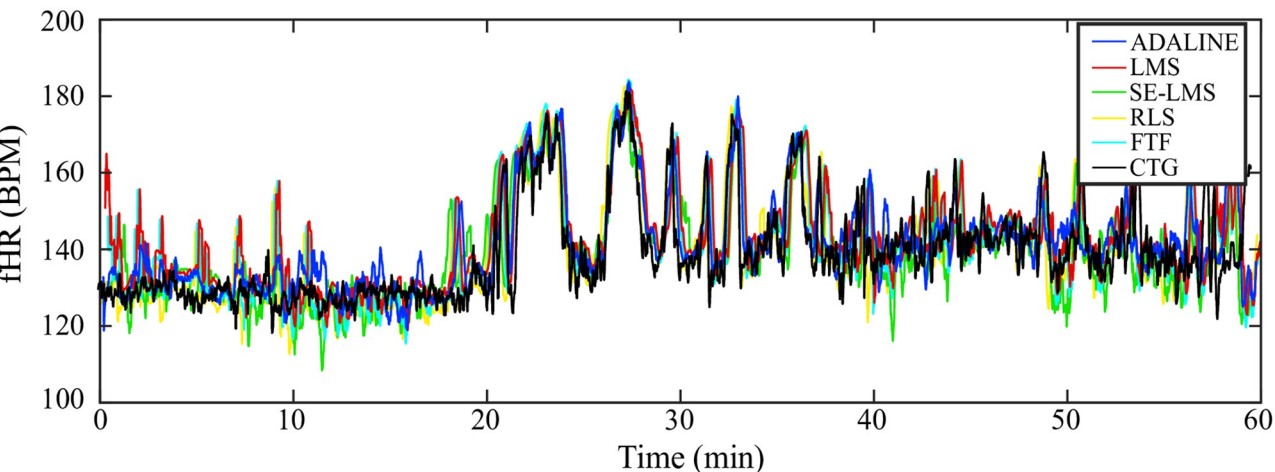

**Fig 7. Comparison of the fHR trace estimated by means of different extraction methods and the CTG reference.** The reference trace from CTG is marked black; the extraction algorithms are marked as follows: ADALINE (blue), LMS (red), SE-LMS (green), RLS (yellow), FTF (cyan).

achieved the worst extraction results. The main difference between the selected recordings is the quality of the scanned aECG signals and the ratio between the maternal and fetal components. Fig 8 demonstrates that some estimates of $mECG^*$ by the ICA algorithm were of low quality, leading to a reduction in the accuracy of fECG extraction. In addition, the morphology of the $mECG^*$ reference signal should correspond to the shape of the parent component in $aECG^*$. However, in the case of r04 and r07 entries, the parent components in both $aECG^*$ signals are bipolar, while the QRS complexes of the $mECG^*$ reference signal have only positive polarity. These differences in the morphology of the maternal component in the individual signals are due to the inaccurate location of the transabdominal leads.

Similarly, the results of the Dataset B from Challenge database were analysed. In Fig 9, we provide the examples of three recordings (a04, a05, a08) that showed outstanding results ($F1 = 100\%$) compared with three recordings (a02, a16, a18) for which the algorithms performed poorly ($F1 < 50\%$). There is a significant difference in the quality between these groups of recordings. The first group (a04, a05, a08) the ratio between the fetal and maternal component is favorable—in some channels, they are of the same magnitude. Contrary, in the latter group (a02, a16, a18), the magnitude of the maternal component is several times higher than the fetal one, which is mostly indistinguishable by the naked eye.

To verify the performance of the proposed hybrid system, we proceeded to compare our results with the results reported in other publicly available professional publications dealing also with the extraction of fECG from aECG using various extraction techniques. Comparing the results of individual studies is relatively difficult, because not all authors use the same data sets and statistical parameters to test the quality of extracted signals. A summary of the methods selected for objective comparison of the achieved results is given in Table 5. Although we selected the results that were tested on same databases (Challenge 2013 and ADFECGDB), the data used for the verification differ. For example, in case of Challenge 2013 database, some authors use all 75 records, whereas some used only part of them. This fact is considered in Table 5 by using the '*' symbol (i.e. Challenge 2013* is dataset that does not include all of the signals). The ADFECGDB initially contained 5 recordings (r01, r04, r07, r08, and r10). However, some authors tested also tested the remaining recordings (r01–r10) that were published later in [30]. Therefore, Table 5 includes both differentiated as "ADFECGDB (5)" and "ADFECGDB", respectively.

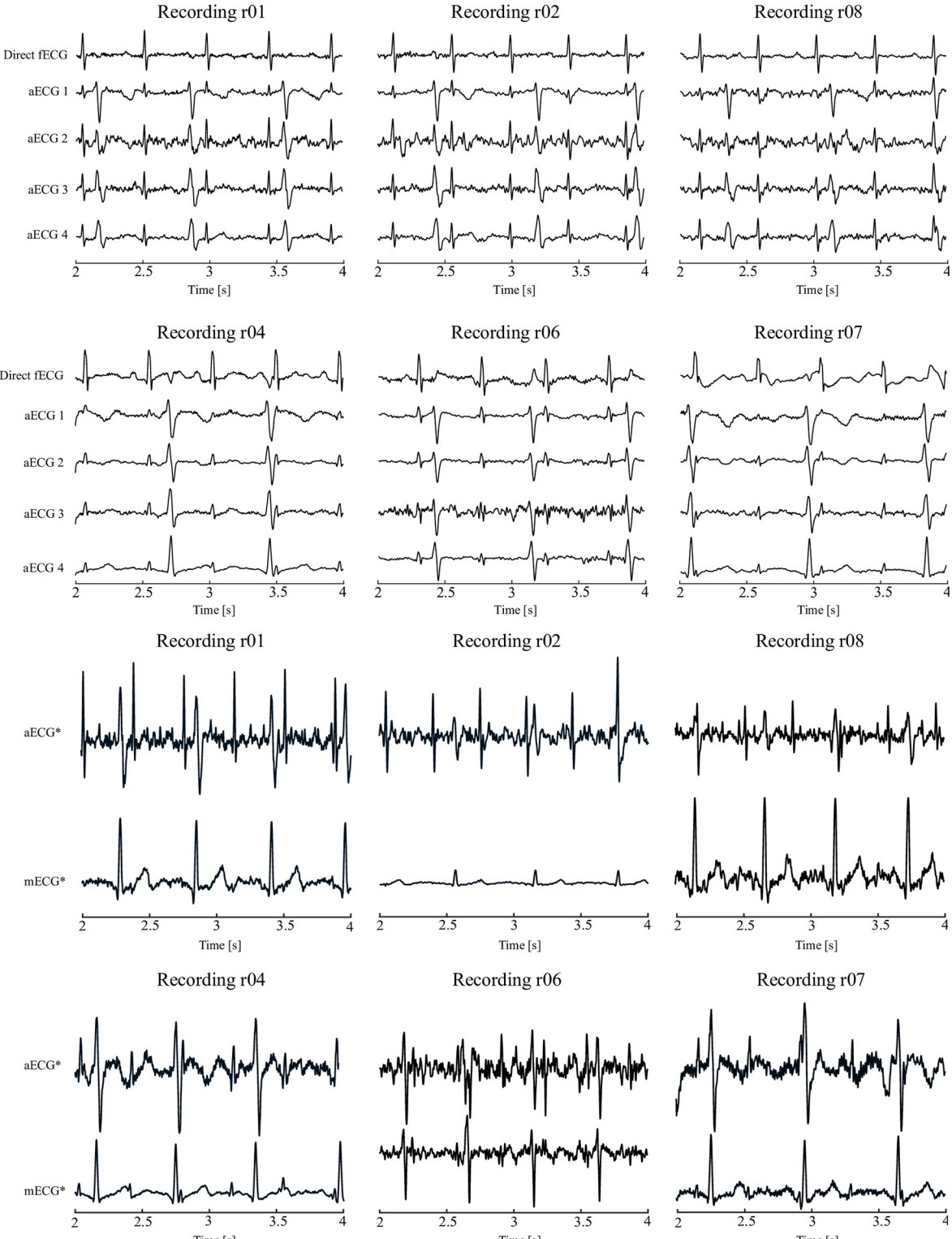

**Fig 8. Examples of aECG signals from the ADFECGDB database.** Three of high quality (r01, r02, r08) and three low quality (r04, r06, r07) along with corresponding *aECG** and *mECG** signals estimated using ICA.

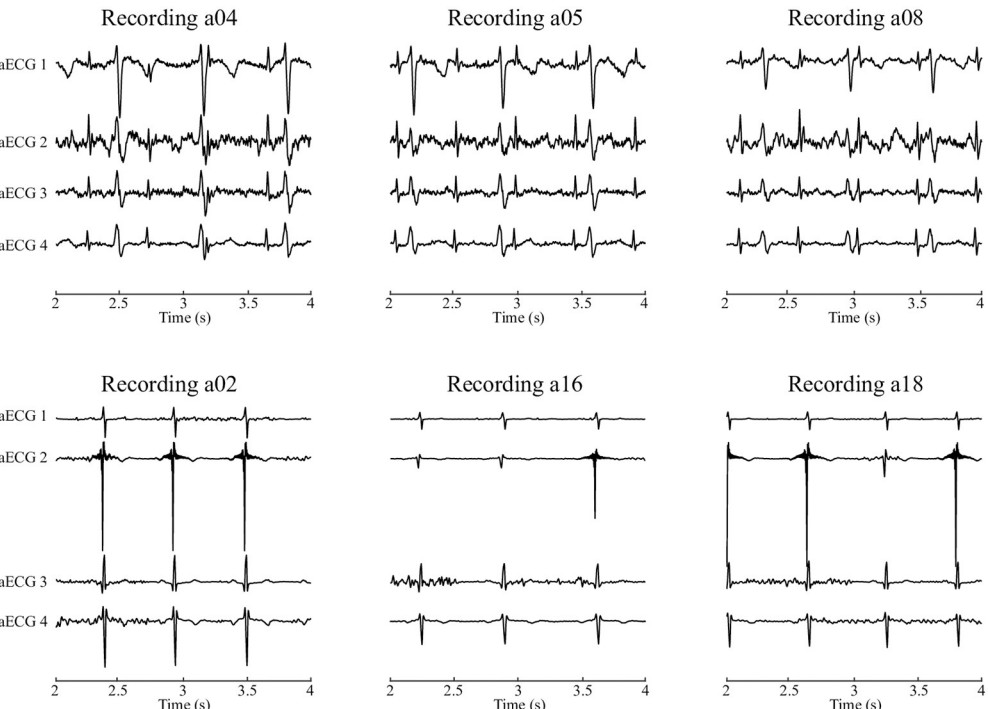

**Fig 9. Examples of aECG signals from the challenge 2013 database.** Three of high quality (a04, a05, a08) and three of low quality (a02, a16, a18).

The most significant results achieved among the recent literature can be described as follows:

- In [4], the authors introduced a combination of the ICA method and an extended Kalman filter. The proposed method was verified on total of 175 min (Challenge data where they disregarded recording 'a54') and approximately 470 min of own data obtained at the gynecology and obstetrics clinic of the University Hospital in Leipzig). The resulting extracted signals were evaluated using statistical parameters Se, PPV, and F1. However, their statistical evaluation (or its summary) is questionable since the authors represented the results only using the values of mean±standard deviation and, for example, their results in terms of F1 parameter for EKF algorithm are as follows: 97.3±10.8 (Challenge data) and 85.4±23.5 (their own data). This repeats through all results. Therefore, a proper comparison of the results is impossible in this case.

- The authors in [33] presented a fECG signal extraction system based on a combination of compressive sensing (CS) and the ICA method. The method was verified on 5 records from the ADFECGDB database and 175 records from the Challenge 2013 database. The parameters Se, PPV, and F1 were used to statistically evaluate the results. The average value of the F1 parameter was 92.20% (ADFECGDB) and 77.50% (Challenge Data), which are very similar to the results obtained in our study for both tested datasets. The authors also analyzed the distribution among the signals of the Challenge dataset and obtained a minimum value for Se parameter equal to 15% (signal a18) and a maximum value 100% (signal a32), while PPV ranged from 21% up to 99%. Moreover, the authors state that the proposed method fails on some abdominal signals (e.g. a02, a09, a18, or a29) of the Challenge dataset due to the poor quality of the signals.

**Table 5. Comparison of the proposed algorithms with other authors.**

| Authors, year, source | Methods | Database | Statistical evaluation | | |
|---|---|---|---|---|---|
| | | | Se(%) | PPV(%) | F1(%) |
| K. Barnova et. al. (2020, 2021), [10, 41] | ICA-RLS-EMD | Challenge 2013* | 81.79 | 87.16 | 84.08 |
| | | Challenge 2013 | 78.00 | 77.00 | 77.50 |
| | ICA-FTF-CEEMDAN | FECGDARHA | 95.33 | 96.40 | 95.86 |
| | | Challenge 2013* | 82.06 | 87.90 | 84.62 |
| R. Li et. al. (2017), [34] | SAVER | ADFECGDB (5) | - | - | 99.36 |
| | ds-AMLMS | | | | 99.55 |
| | ds-AMESN | | | | 99.00 |
| | ds-TSEKF | | | | 96.85 |
| | ds-TSPCA | | | | 98.52 |
| | SAVER | Challenge 2013 | - | - | 87.93 |
| | ds-AMLMS | | | | 72.77 |
| | ds-AMESN | | | | 72.04 |
| | ds-TSEKF | | | | 86.67 |
| | ds-TSPCA | | | | 91.72 |
| A. Krupa et. al. (2021), [37] | FrFT-WT | Challenge 2013(Only 12 rec.) | 95.18 | 97.11 | 96.11 |
| W. Zhong et. al. (2018), [38] | Tree-search method | ADFECGDB (5) | 91.95 | 92.76 | 92.34 |
| E. Castillo et. al. (2018), [39] | WT-CT | ADFECGDB (5) | 98.40 | 98.86 | 98.63 |
| | | Challenge 2013* | 97.93 | 99.11 | 98.52 |
| R. Jaros et. al. (2019), [12] | ICA-RLS-WT | ADFECGDB | 89.70 | 92.41 | 90.99 |
| | | Challenge 2013 | 72.59 | 81.34 | 75.67 |
| | ICA-ANFIS-WT | ADFECGDB | 71.05 | 76.29 | 73.29 |
| | | Challenge 2013 | 67.09 | 72.76 | 69.44 |
| W. Zhong et. al. (2019), [40] | RCED-Net | ADFECGDB (5) | 96.06 | 92.25 | 94.10 |
| | | Challenge 2013* | 92.60 | 94.68 | 93.62 |
| Proposed algorithms | ICA-LMS | ADFECGDB | 92.02 | 93.89 | 92.88 |
| | ICA-SeLMS | | 90.92 | 92.88 | 91.80 |
| | ICA-ADALINE | | 94.85 | 93.30 | 94.06 |
| | ICA-RLS | | 94.48 | 93.20 | 93.79 |
| | ICA-FTF | | 84.34 | 91.46 | 87.27 |
| | ICA-LMS | Challenge 2013/Challenge 2013* | 73.15/89.41 | 75.46/90.42 | 73.77/89.19 |
| | ICA-SeLMS | | 75.22/86.50 | 75.52/85.09 | 74.58/84.53 |
| | ICA-ADALINE | | 78.76/93.57 | 76.87/90.87 | 78.81/92.47 |
| | ICA-RLS | | 79.58/90.43 | 77.67/90.34 | 78.45/88.08 |
| | ICA-FTF | | 68.69/77.91 | 64.53/75.79 | 66.22/74.28 |

*Challenge 2013—statistics conducted using 26 selected recordings from Challenge 2013 Training Set A,
ADFECGDB (5)—evaluation using 5 recordings from ADFECGDB dataset (r01, r04, r07, r08, r10).

- In [34], the authors test the SAVER (Smart Adaptive Ecg Recognition) method based on modern techniques of time-frequency analysis and multiple learning. The proposed algorithm uses the short Fourier transform (dsSTFT) and the non-local median algorithm to estimate the maternal and fetal components. The proposed method was tested on two publicly available databases: ADFECGDB(5) and Challenge 2013*. The resulting value of the F1 parameter for the ADFECGDB database is 99.36% and for Challenge 2013 87.93%. To verify the performance of the proposed system, the authors also tested combinations of other

methods proposed, for example, in [32, 35, 36]. The results of the F1 parameter for all experiments are given in Table 5.

- Krupa et al. [37] proposed fECG extraction in the time-frequency domain using FrFT-WT algorithm combining fractional Fourier transform (FrFT) and WT. The FrFT was used to suppress maternal component while WT to reduce further interference. When testing on records from Challenge 2013 dataset (only on 12 selected recordings), ACC = 92.68% was achieved and when testing on own real records ACC = 96.98%.

- In [38], the authors introduced a tree-search method for fHR monitoring from single channel abdominal ECG recordings. The proposed method is composed of three main stages: a preprocessing stage, a new tree-search methodology for detecting fetal QRS complexes, and a final stage for false positive and false negative correction. Two databases were used to illustrate the efficiency of the proposed method (Daisy and ADFECGDB). For the data from the ADFECGDB(5) database, the average value of the parameter F1 was 92.34%, similarly as in the work presented herein.

- The combination of wavelet transform (WT) and clustering-based technique (CT) was tested in [39]. The effectiveness of the method was tested on ADFECGDB(5) and 26 records from Challenge 2013*. Moreover, the authors excluded some of the aECG abdominal signals where fetal heart beats were not detectable, and signals that were affected by severe noise. These excluded signals were r04 Ab-1, r07 Ab-1, and r10 Ab-3. The extraction results were evaluated according to statistical parameters F1, PPV, Se, and ACC. The authors state that the combination of WT-CT algorithms reached the average value of the parameter F1 98.63% for data from ADFECGDB and 98.52% for data from Challenge 2013, which is similar to our results.

- The authors in [12] presented a new hybrid algorithm using a combination of ICA-ANFIS-WT and ICA-RLS-WT methods. The study was performed on data from clinical practice (extended database ADFECGDB—12 records and database Physionet Challenge 2013—25 records). The extraction results were evaluated using statistical parameters Se, PPV, and F1. The mean F1 score obtained for the ADFECGDB data was 73.29% for the ICA-ANFIS-WT combination and 90.99% for the ICA-RLS-WT. For the data from Challenge 2013, the value of the parameter F1 was 69.44% (ICA-ANFIS-WT) and 75.67% (ICA-RLS-WT).

- In [40], the authors proposed a new approach to fECG extraction using a deep learning strategy from a single channel aECG record based on residual coder-decoder (RCED-Net) convolution networks. The authors tested their extraction system on two databases containing real data: ADFECGDB(5) and Challenge 2013 (75 records). The resulting extracted signals were further tested using the statistical parameters Se (%), PPV (%), and F1 (%). The study achieved 94.10% average accuracy of R-peak detection according to the F1 parameter for the ADFECGDB database and 93.62% for Challenge 2013 data.

- Barnova et al. introduced several algorithms, e.g. [10, 11, 41] combining EMD-based methods with different adaptive algorithms (such as RLS and FTF). The efficacy of the ICA-RLS-EMD algorithm introduced in [10] was tested on Challenge dataset, reaching Se = 81.79%, PPV = 87.16%, and F1 = 84.08%. In [11], the authors introduced alternative of this method by replacing the EMD element with Ensemble EMD (EEMD). Finally, the authors introduced the ICA-FTF-CEEMDAN method, where the last step replaced by complementary EEMD with adaptive noise (CEEMDAN). The authors tested the proposed methods on the Fetal Electrocardiograms, Direct and Abdominal with Reference Heartbeats Annotations database and achieved ACC higher than 80% for 11 out of 12 recordings. In contrast, when

testing on Challenge 2013 database, accuracy higher than 80% was achieved for 17 out of (only) 25 selected recordings. The best results were achieved for ICA-FTF-CEEMDAN method introduced in [41], which was tested on the Fetal Electrocardiograms, Direct and Abdominal with Reference Heartbeats Annotations (FECGDARHA) database (average values of ACC 92.98%). On the Challenge 2013 database, the method achieved average values of ACC = 78.47%. The main drawbacks of these methods mentioned are their computational complexity and the need to individually set the parameters of the RLS/FTF and EEMD-based algorithms in each use. This makes the method very hardly implementable into clinical practice.

This study focused on finding the optimal setting of filtration parameters for selected adaptive algorithms. The subject of further research will be the testing of new hybrid algorithms, which are very promising and whose core is adaptive filtering. In addition to monitoring fHR, these new methods also have the potential for deeper morphological analysis. In the future, this could enable early detection of fetal hypoxia and during the childbirth, which could lead to reduced number of cesarean sections, but also for monitoring and diagnosis through the pregnancy. For these purposes, for example, the analysis of the ST segment could be used. Non-invasive variant of the method can be used not only to predict potential risk of fetal hypoxia, but also for other diagnosis regarding fetal health state or various pathologies through, for example, analysis of the length of the QT interval. Our team has already introduced a pilot study of this issue in [7].

Finally, the optimal placement of electrodes in NI-fECG measurement still remains unstandardised. It should be noted that the electrode deployment differs for various gestational age, fetal position, and stage of pregnancy. Similarly, the optimal filter settings also depend on these factors, as they directly affect the quality of the sensed aECG signal. The future research should therefore thoroughly investigate this area of research.

## Conclusion

In this paper, we introduced a novel approach for optimization and training of hybrid systems for fECG signal extraction in real signals from clinical practice. Our proposed approach has the potential to emerge as a very useful complimentary method to support the conventional in the field of obstetrics and gynaecology. Several different adaptive algorithms were included in the tests, namely ADALINE, LMS, Sign-Error LMS, RLS, and FLF algorithms. The results showed that our method improves the extraction of highly accurate fECG signals, especially using the ADALINE adaptive block (average accuracy assessed by F1 = 94.85% and 92.47% for ADFECGDB and Challenge 2013 datasets, respectively). The approach is also associated with negligible effect on their morphology thus enhances the clinical diagnostic capability of the NI-fECG method. Moreover, it is significantly less susceptible to the effects of electrode placement and system configuration on signal quality. This was verified on the data from publicly available databases and on the measurements in clinical practice. Therefore, our approach ultimately paves the way for more accurate detection and estimation of fetal hypoxic conditions in a non-invasive fashion.

## Author Contributions

**Conceptualization:** Radana Kahankova, Radek Martinek.

**Data curation:** Radana Kahankova.

**Formal analysis:** Radana Kahankova, Martina Mikolasova, Radek Martinek.

**Investigation:** Radana Kahankova, Martina Mikolasova.

**Methodology:** Radana Kahankova.

**Project administration:** Radek Martinek.

**Resources:** Radek Martinek.

**Software:** Radana Kahankova.

**Supervision:** Radana Kahankova.

**Validation:** Radana Kahankova, Martina Mikolasova.

**Visualization:** Radana Kahankova, Martina Mikolasova.

**Writing – original draft:** Radana Kahankova, Martina Mikolasova.

**Writing – review & editing:** Radana Kahankova, Martina Mikolasova, Radek Martinek.

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
