## [Decision Letter · Decision Letter 0]

14 Feb 2022

PONE-D-21-24207Optimization of Adaptive Filter Control Parameters for Non-Invasive Fetal Electrocardiogram ExtractionPLOS ONE

Dear Dr. Kahankova,

Thank you for submitting your manuscript to PLOS ONE. After careful consideration, we feel that it has merit but does not fully meet PLOS ONE’s publication criteria as it currently stands. Therefore, we invite you to submit a revised version of the manuscript that addresses the points raised during the review process.

We look forward to receiving your revised manuscript.

Kind regards,

Ardashir Mohammadzadeh, Phd

Academic Editor

PLOS ONE

Journal Requirements:

"R.M. and R.J. European Regional Development Fund in the Research Centre of Advanced Mechatronic Systems project through the Operational Programme Research, Development and Education under Project CZ.02.1.01/0.0/0.0/16_019/0000867

M.M. Ministry of Education of the Czech Republic under Project SP2021/32."

3. Please ensure that you refer to Figure 9 in your text as, if accepted, production will need this reference to link the reader to the figure.

4. We note you have included a table to which you do not refer in the text of your manuscript. Please ensure that you refer to Tables 1, 2, 3, 4, and 5 in your text; if accepted, production will need this reference to link the reader to the Table.

Reviewers' comments:

Reviewer's Responses to Questions

**Comments to the Author**

1. Is the manuscript technically sound, and do the data support the conclusions?

Reviewer #1: Yes

Reviewer #2: Yes

2. Has the statistical analysis been performed appropriately and rigorously? 

Reviewer #1: Yes

Reviewer #2: Yes

3. Have the authors made all data underlying the findings in their manuscript fully available?

Reviewer #1: Yes

Reviewer #2: Yes

4. Is the manuscript presented in an intelligible fashion and written in standard English?

Reviewer #1: Yes

Reviewer #2: Yes

5. Review Comments to the Author

Reviewer #1: This paper is focused on the design, implementation and verification of a novel method for the optimization of the control parameters of different hybrid systems used for non-invasive fetal electrocardiogram (fECG) extraction. This paper is generally well written with sufficient analysis. It can be accepted if the following concerns can be addressed.

1. Please ensure all the variables have been defined clearly in the paper.

2. The parameter tuning is critical to the performance of the discussed method, so more details should be given on how to select the parameters.

3. Least squares-based methods such as RLS and LMS have been widely used for system parameter identification, e.g., DOI: 10.1109/JESTPE.2021.3098836; DOI: 10.1109/TTE.2020.2994543. Such works can be included for brief discussion.

4. Least squares-based identification is vulnerable to the noises on system input. Noise-compensated versions of them have been proposed in the literature for noise-immune and unbiased identification, e.g., DOI: 10.1109/TIE.2019.2962429; DOI: 10.1109/TIE.2021.3063968. Such works can be mentioned for completeness and maybe as a future work.

5. The conclusions can be further improved. Primary findings with statistical results should be given.

Reviewer #2: This article raises a very interesting topic and will definitely be discussed further in the future. And after applying a minor change can be published

1- More literature needs to be surveyed.

2- Discussion is short. Please express it more detailed.

6. PLOS authors have the option to publish the peer review history of their article (what does this mean?). If published, this will include your full peer review and any attached files.

Reviewer #1: No

Reviewer #2: No

---

## [Author Response · Author response to Decision Letter 0]

21 Feb 2022

Responses to Editor:

Comment 1. Please ensure that you refer to Figure 9 in your text as, if accepted, production will need this reference to link the reader to the figure.

Authors’ action: We added reference to Fig. 6 (lines 279, 294) and Fig. 9 (line 417).

Comment 2. We note you have included a table to which you do not refer in the text of your manuscript. Please ensure that you refer to Tables 1, 2, 3, 4, and 5 in your text; if accepted, production will need this reference to link the reader to the Table.

Authors’ action: We went through the manuscript, all tables are mentioned: Tab1 (lines 102 and 117), Tab2 (line 278), Tab 3 (lines 326, 328, 338, 345), Tab4 (line 363), Tab5 (lines 431, 434, 438, 474) 

 

Reviewer #1

This paper is focused on the design, implementation and verification of a novel method for the optimization of the control parameters of different hybrid systems used for non-invasive fetal electrocardiogram (fECG) extraction. This paper is generally well written with sufficient analysis. It can be accepted if the following concerns can be addressed.

Comment 1: Please ensure all the variables have been defined clearly in the paper.

Authors’ answer: Thank you, for your suggestion. We went through the text and corrected parts where the variables were not sufficiently described.

Authors’ action: We found the cases where the variables were not sufficiently defined and corrected them.

Comment 2: The parameter tuning is critical to the performance of the discussed method, so more details should be given on how to select the parameters.

Authors’ answer: Thank you, for your suggestion, this will be added.

Authors’ action: We added more details regarding selection of the parameters to be tuned.

Comment 3: Least squares-based methods such as RLS and LMS have been widely used for system parameter identification, e.g., DOI: 10.1109/JESTPE.2021.3098836; DOI: 10.1109/TTE.2020.2994543. Such works can be included for brief discussion.

Authors’ answer: Thank you, for your suggestion. We mentioned system parameter identification as one of the use cases of LMS algorithms. The works are from quite different area not related to biosignals, so we did not make extended discussion of them.

Authors’ action: We mentioned system parameter identification as one of the use cases of LMS algorithms (line 133). We included the mentioned works as references.

Comment 4: Least squares-based identification is vulnerable to the noises on system input. Noise-compensated versions of them have been proposed in the literature for noise-immune and unbiased identification, e.g., DOI: 10.1109/TIE.2019.2962429; DOI: 10.1109/TIE.2021.3063968. Such works can be mentioned for completeness and maybe as a future work.

Authors’ answer: Thank you, for your suggestion. We mentioned this in the introduction but since the mentioned works and the approach is from a completely different field that is not related to proposed study, we did not include it into the future work in further discussions.

Authors’ action: We added this information and reference to the introduction.

Comment 5: The conclusions can be further improved. Primary findings with statistical results should be given.

Authors’ answer: Thank you, for your suggestion. Indeed, the conclusion was not sufficient. 

Authors’ action: We expanded the conclusion section.

Reviewer #2 

This article raises a very interesting topic and will definitely be discussed further in the future. And after applying a minor change can be published

Comment 1: More literature needs to be surveyed.

Authors’ answer: Thank you, for your suggestion, we extended the literature survey for the latest research outcomes and added them to the introduction and comparison of the results.

Authors’ action: The literature survey was extended.

Comment 2: Discussion is short. Please express it more detailed.

Authors’ answer: Thank you, for your suggestion, the discussion was not structured right and seemed short.

Authors’ action: We revised the discussion section.

---

## [Decision Letter · Decision Letter 1]

21 Mar 2022

PONE-D-21-24207R1Optimization of Adaptive Filter Control Parameters for Non-Invasive Fetal Electrocardiogram ExtractionPLOS ONE

Dear Dr. Kahankova,

Thank you for submitting your manuscript to PLOS ONE. After careful consideration, we feel that it has merit but does not fully meet PLOS ONE’s publication criteria as it currently stands. Therefore, we invite you to submit a revised version of the manuscript that addresses the points raised during the review process.

We look forward to receiving your revised manuscript.

Kind regards,

Ardashir Mohammadzadeh, Phd

Academic Editor

PLOS ONE

Journal Requirements:

Additional Editor Comments:

One of the reviewers has some concern about the novelties of paper; please revise the paper according the comments.

Reviewers' comments:

Reviewer's Responses to Questions

**Comments to the Author**

1. If the authors have adequately addressed your comments raised in a previous round of review and you feel that this manuscript is now acceptable for publication, you may indicate that here to bypass the “Comments to the Author” section, enter your conflict of interest statement in the “Confidential to Editor” section, and submit your "Accept" recommendation.

Reviewer #2: All comments have been addressed

2. Is the manuscript technically sound, and do the data support the conclusions?

Reviewer #2: Yes

3. Has the statistical analysis been performed appropriately and rigorously? 

Reviewer #2: Yes

4. Have the authors made all data underlying the findings in their manuscript fully available?

Reviewer #2: Yes

5. Is the manuscript presented in an intelligible fashion and written in standard English?

Reviewer #2: Yes

6. Review Comments to the Author

Reviewer #2: What is the difference between your work and the method presented in the following article?

Katerina Barnova, Radek Martinek, Rene Jaros, Radana Kahankova, Khosrow Behbehani, Vaclav Snasel,

System for adaptive extraction of non-invasive fetal electrocardiogram,

Applied Soft Computing,

Volume 113, Part B,

2021,

107940,

ISSN 1568-4946,

https://doi.org/10.1016/j.asoc.2021.107940.

7. PLOS authors have the option to publish the peer review history of their article (what does this mean?). If published, this will include your full peer review and any attached files.

Reviewer #2: No

---

## [Author Response · Author response to Decision Letter 1]

27 Mar 2022

Responses to Editor:

Comment 1. One of the reviewers has some concern about the novelties of paper; please revise the paper according the comments.

Authors’ action: We revised the paper according the comments.

Reviewer #2: What is the difference between your work and the method presented in the following article?

Katerina Barnova et al. System for adaptive extraction of non-invasive fetal electrocardiogram, Applied Soft Computing, Volume 113, Part B, 2021, 107940, ISSN 1568-4946, https://doi.org/10.1016/j.asoc.2021.107940.

Authors’ answer: In the proposed work, we tested and optimized in total of 5 algorithms in hybrid fECG extraction system: ADALINE, LMS, SE-LMS, RLS and FTF. The combinations of ICA-ADALINE and ICA-SELMS and have not been used in the literature before. In the paper of Barnova you mentioned, the focus is fully on the ICA-FTF-CEEMDA extraction system. However, in our results, the FTF algorithm is achieving poorest results when used alone, i.e. without final processing using CEEMDA (method who’s settings also need to be optimized). In our paper, the ADALINE method is most powerful when used alone. Therefore, it could be further improved when additional processing enhancing the fetal R peaks was used (such as CEEMDA method). Further, Barnova in her paper admits that the limitation of her work is the need for optimal parameter setting of both FTF and CEEMDA algorithms (see table 8 summary in her paper). Contrary to paper of Barnova et al., we did not focus only on one method but on finding a way to optimize them all so they function well even without additional postprocessing methods – i.e. optimizing them for fECG extraction (as stated in the title of the article).

Authors’ action: We added the information regarding this paper to the manuscript and highlighted the limitations.

---

## [Editor Report · Decision Letter 2]

29 Mar 2022

Optimization of Adaptive Filter Control Parameters for Non-Invasive Fetal Electrocardiogram Extraction

PONE-D-21-24207R2

Dear Dr. Kahankova,

We’re pleased to inform you that your manuscript has been judged scientifically suitable for publication and will be formally accepted for publication once it meets all outstanding technical requirements.

Kind regards,

Ardashir Mohammadzadeh, Phd

Academic Editor

PLOS ONE
---

## [Editor Report · Acceptance letter]

1 Apr 2022

PONE-D-21-24207R2 

Optimization of Adaptive Filter Control Parameters for Non-Invasive Fetal Electrocardiogram Extraction 

Dear Dr. Kahankova:

I'm pleased to inform you that your manuscript has been deemed suitable for publication in PLOS ONE. Congratulations! Your manuscript is now with our production department. 

Kind regards, 

on behalf of

Dr. Ardashir Mohammadzadeh 

Academic Editor

PLOS ONE